# A Deep Learning-Based Semantic Segmentation Model Using MCNN and Attention Layer for Human Activity Recognition

**DOI:** 10.3390/s23042278

**Published:** 2023-02-17

**Authors:** Sang-hyub Lee, Deok-Won Lee, Mun Sang Kim

**Affiliations:** School of Integrated Technology, Gwangju Institute of Science and Technology, Gwangju 61005, Republic of Korea

**Keywords:** human activity recognition, transitional activities, deep learning, accelerometer sensor, attention layer, semantic segmentation

## Abstract

With the development of wearable devices such as smartwatches, several studies have been conducted on the recognition of various human activities. Various types of data are used, e.g., acceleration data collected using an inertial measurement unit sensor. Most scholars segmented the entire timeseries data with a fixed window size before performing recognition. However, this approach has limitations in performance because the execution time of the human activity is usually unknown. Therefore, there have been many attempts to solve this problem through the method of activity recognition by sliding the classification window along the time axis. In this study, we propose a method for classifying all frames rather than a window-based recognition method. For implementation, features extracted using multiple convolutional neural networks with different kernel sizes were fused and used. In addition, similar to the convolutional block attention module, an attention layer to each channel and spatial level is applied to improve the model recognition performance. To verify the performance of the proposed model and prove the effectiveness of the proposed method on human activity recognition, evaluation experiments were performed. For comparison, models using various basic deep learning modules and models, in which all frames were classified for recognizing a specific wave in electrocardiography data were applied. As a result, the proposed model reported the best F1-score (over 0.9) for all kinds of target activities compared to other deep learning-based recognition models. Further, for the improvement verification of the proposed CEF method, the proposed method was compared with three types of SW method. As a result, the proposed method reported the 0.154 higher F1-score than SW. In the case of the designed model, the F1-score was higher as much as 0.184.

## 1. Introduction

### 1.1. Research Background

Various issues about the safety and health of the elderly have emerged in our aging society. Studies are being conducted to prevent these issues. Particularly, the awareness of daily activity is becoming more important because it is directly related to the health of the elderly. Owing to an aging society, the elderly population is increasing, but there is a limit to the manpower that can take care of them; thus, a technology that can replace the elderly care manpower is required. For this reason, with the recent development of wearable devices and deep learning (DL)-based artificial intelligence technology, human activity recognition (HAR) is being employed to recognize what people are doing through a series of data over time.

HAR is a technology suitable for the current healthcare field in an aging society. This is because data on perceived human activity can be used in various technological fields, such as human–computer interaction and human–robot interaction (HRI) [1]. By fusion with the internet of things (IoT) technology or timeseries sensor data, it is possible to propose appropriate services for various targets. For example, a mobile robot that can operate in an indoor environment can provide appropriate and proactive new services such as medication recognition for the elderly considering recognized activities. In addition, HAR can generate significant information for implementing a home care system. It is crucial to quickly and accurately recognize issues directly related to diseases or health, such as falls, in the time domain. In this sense, the HAR technology can be of great significance for distributing a monitoring system in a real environment [2].

There are three typical types of data used for HAR [3]. The first one is biosignal data such as electroencephalography, electromyography, and electrocardiography (ECG). Such data cannot be collected easily because the data collection requires specific equipment, including an electrode for recording electrical signals. The second one is behavior-sensing data such as the type of image obtained from an RGB-D sensor. They provide lots of useful information for HAR in the form of the original image and the skeleton type extracted from the depth image. However, because there are issues such as privacy invasion, it is unsuitable for application to people’s home environments. In addition, an RGB-D sensor has a coverage limitation in that the target must be located in the field of view of the sensor during data collection, and there should be no occlusions that compromise the data quality. These issues significantly influence the recognition of the behavior of many objects. Therefore, it is unsuitable for an elderly home care or monitoring system in daily life. The last type is activity-sensing data from an inertial measurement unit (IMU) comprising an accelerometer and a gyroscope. They have high usability according to the development of wearable devices, and the privacy issue is less severe. Moreover, because data can be collected from the sensor itself or using specific anchors with signal communication, coverage limitations are less impactful for the application than RGB-D sensors. Most IMU data are obtained from a wearable sensor attached to the body and have high scalability because they can easily be fused with other sensor data that have timeseries characteristics. For example, if IMU data are combined with indoor localization technology such as ultra-wide-band (UWB) sensing, not only the information necessary for behavior recognition but also context information such as the target position can be obtained. Such sensor fusion improves the performance of behavior recognition and can be the key to HRI or IoT technology. For that reason, IMU data are the most appropriate data type for HAR in daily life [4]. Therefore, a new HAR method using IMU data, particularly acceleration data, is proposed in this study.

According to the development of DL, many scholars have performed HAR using timeseries acceleration data collected with wearable sensors. An elaborate HAR can be achieved by detecting the start and end points of the target activity in timeseries data that include single or multiple activities. Most scholars first performed a segmentation task on the entire timeseries data into the optimal size for classifying target activities [5] and then classified each segment. However, this approach has limitations in performance because human activities are not standardized for each person [6]. In detail, the fixed-size window (FSW) method, shown on the left of Figure 1, cannot cover properly when the target activity execution time is larger than the window size. In addition, the case that there are multiple activities in a single window causes low classification accuracy.

To tackle the above issue, the sliding window (SW) method, shown on the right of Figure 1, has been used recently. The classification window moves along the time axis considering the size of the overlapping area, and classification is performed for every step. However, as with the FSW method, the SW method still has an issue with determining the optimal window size and overlapping area. In the results of several studies, different optimal sizes have been reported for different datasets. In addition, there remains a generalization problem for the recognition performance of the obtained optimal size. In particular, because the duration of human activities is not constant, there are limitations in accurately classifying behavior even using SW.

To solve the above problems, a method of classification for every frame (CEF) in timeseries acceleration data is proposed in this study. The proposed method is similar to the segmentation method presented in fields of two-dimensional (2D) image recognition named semantic segmentation. In addition, a DL-based new architecture is designed for conducting semantic segmentation on three-axis acceleration data.

### 1.2. Related Work

Many scholars have conducted recognition-based model development studies for HAR using convolutional neural networks (CNNs) or recurrent neural networks (RNNs). Particularly, most scholars have developed a HAR model using CNN. In [7], a one-dimensional CNN (1D-CNN)-based model for HAR was proposed. For the input, the acceleration data from multiple IMU sensors attached at different positions of body parts were used. The proposed model applies multiple CNN (MCNN) and pooling layers to each sensor data separately. In addition, extracted features are concatenated and used to predict the segment class. The proposed classification model predicts which behavior each segment corresponds to. For the evaluation, three public datasets of human activity presented in [8,9,10] were used. The window sizes were 0.72, 3, and 1 s for each dataset, and the size of the overlapping area of SW was set to 50%, 78%, and 99% of the segmented window sizes, respectively. As the result, the accuracy was 92.22%, 93.68%, and 70.80%, respectively. The authors of [11] proposed a 1D-CNN-based model for recognition. The proposed model extracts meaningful features by capturing local dependencies and scale invariance of timeseries activity data acquired by IMU. Similar to [7], the recognition model included a channel-wise 1D-CNN layer (applying 1D-CNN layers to x, y, and z channels) and a pooling layer. Moreover, the window size and the size of the overlapping area were 64 and 50%, respectively. For the evaluation, the public human activity datasets using IMU presented in [8,12,13] were used. The accuracy for each dataset was 76.83%, 88.19%, and 96.88%. The authors of [14] proposed a HAR model using 1D-CNN and a conquer-based classifier. First, the proposed model recognized activity as static (sit, stand, and lay) or dynamic (walking, walking upstairs, and walking downstairs) activity using binary classification. Then, two three-class classifiers were implemented to predict the class of each FSW. Finally, test data sharpening was adopted to improve the HAR performance. The window size and size of the overlapping area were 500 and 250 ms, respectively. The proposed model was evaluated on two public datasets presented in [8,15]. As a result, the accuracy for each dataset was 94.2% and 97.62%. In [16], multiple DL architectures, including deep feed-forward neural network, CNN, long short-term memory (LSTM), and bidirectional LSTM, were implemented for HAR. The recognition models were evaluated on three public datasets [8,9,17] using different window sizes (1, 5.12, and 1 s) and the size of the overlapping area (50%, 78%, and 50%). As a result, the bidirectional LSTM showed the best performance on the three datasets, with F1-scores of 0.929 for [8], 0.745 for [9], and 0.76 for [17]. The authors of [18] used stacked LSTM modules for HAR. Further, the SW method was implemented with a window size of 10 s, and 90% of the window size was set as the size of the overlapping area. As a result, an accuracy of 94% was achieved for the public dataset presented in [19]. The authors of [20] proposed a HAR model using bidirectional LSTM modules with a residual connection. For the classifier implementation, the input data have FSW with a 2.56-s segment. In addition, the size of the overlapping area of SW was 50% of the window size. Notably, the residual connection made the model optimization much easier than the original structure because the gradient values used in the learning process could be spread to the layers more directly through the residual connection. As a result, the F1-score for two public datasets presented in [8,15] was 0.905 and 0.935. The key characteristics of introduced previous works are shown in Table 1.

Owing to the aforementioned studies, the performance of HAR using SW has been improved by applying the DL technology. However, some issues remain. First, the size of the fixed window differs for each proposed model. Although the same model is used, the optimal size of the fixed window and the size of the overlapping area of SW differ for different datasets. This means that the SW using a fixed size is difficult to generalize. In other words, when the SW method is implemented on data collected in different environments, the error rate could be increased. Second, as mentioned in Section 1, the duration of human activity is usually variable. Therefore, the performance of a proposed HAR model could vary according to the size of the window used as input for the recognition model. Similarly, the size of the overlapping area of SW also affects the performance. The authors of [19] found that window size is a key parameter for improving recognition accuracy. A too-small window size could not include the entire activity, and a too-large window size could be a reason for classification error. In [6], the window size of SW significantly influenced the recognition performance. In addition, the authors mentioned that the optimal window size is hard to predefine because of the inconstancy of the type and duration of human activity. Further, a predefined optimal window size could differ for various unseen activities. Therefore, the SW method finds it difficult to handle the various activities. Finally, in the process of learning SW, the issue of determining the label of each segment remains. Most SW studies set the label of each segment as the class corresponding to the most part in the segment or the last frame of the segment to improve the training performance. This means that more than one activity could exist in a single window, and the proportion occupied by each class may be biased. This can degrade the performance of a recognition model on real data and prevent accurate classification. To tackle these issues, the CEF-based semantic segmentation method is proposed in this study.

The remainder of this article is structured as follows. Section 2 describes a DL model made up of stacking MCNN and the designed attention layer. In addition, a new dataset comprising only types of transition activities is described in Section 2. Section 3 describes the evaluation results of the proposed method compared with the basic DL models and previous work that conducted semantic segmentation of ECG data. Further, the performance of CEF was evaluated by comparing it with the existing SW method. Section 4 analyzes and discusses the experimental results. Finally, the conclusion and future work are presented in Section 5.

## 2. Materials and Methods

### 2.1. CEF Using DL Model

To implement CEF, the method of semantic segmentation was adopted. Semantic segmentation was proposed for 2D image segmentation; it usually means detecting pixels corresponding to the target in an input image. Similar to image segmentation, the semantic segmentation method was adopted for timeseries data in this study. The designed recognition model used MCNN as the feature extraction layer in timeseries data. In addition, an attention layer similar to CBAM was designed to improve recognition performance.

#### 2.1.1. Feature Extraction Block Using MCNN

The timeseries data comprises different features along the time axis. In the case of data collected using an IMU sensor, the features correspond to the x, y, and z axes. For the feature extraction of timeseries data, 1D-CNN is appropriate because the kernel only moves along the time axis. In addition, the convolutional kernel extracts the features using data of a certain size in the time range. Therefore, 1D-CNN can operate as a local feature extractor in timeseries data. However, the definition of optimal kernel size for improving the recognition performance could not be specified, similar to the problem of SW based on a fixed window. Consequently, multiple 1D-CNNs with different kernel sizes were adopted. Multiple features that consider multiple receptive fields with various ranges can be extracted using MCNN. The designed architecture in this study was inspired by the SPP-Net proposed for image classification [21]. The proposed feature extraction layer uses five 1D-CNN layers with different kernel sizes of 5, 10, 20, 50, and 100. Each layer controls the padding size to make the size of the output data to be the same as that of the input. Each kernel performs a convolution operation on the input data and extracts features by sliding along the time axis. In other words, the features of each time point are extracted considering the surrounding data in different ranges. Then, the extracted features are fed into two 1D-CNNs with kernel size 1. Afterward, different features from each 1D-CNN are concatenated along the channel axis. Finally, features are fed into a single 1D-CNN layer with kernel size 1. The last layer not only adjusts the size of features, but also performs the fusion considering meaningful features among features extracted from each kernel with a different size. The detailed shape of the feature extraction block is shown in Figure 2. After every 1D-CNN layer, a batch normalization function is added to prevent overfitting; a layer normalization function is additionally added to prevent the feature values from becoming too large. The proposed feature extraction block is stacked in several steps to make our model deep.

#### 2.1.2. Implementation of the Attention Layer

To improve recognition performance, an attention layer similar to CBAM presented in [22] was implemented in the designed model. In addition, the attention layer integrates multiple features by weighting for each channel and each time step. Further, it can guide the feature extraction block to extract important features as well. The designed attention layer was based on CBAM reported in image recognition; it performs the attention mechanisms for channel and spatial separately. For the channel attention layer of CBAM, an attention score indicating which input channel is more important is generated with a probability distribution. To obtain the channel-level attention map used for calculating the attention score, average pooling and max pooling are applied to the input data in the spatial direction. Then, the attention score is calculated using a sigmoid function after a feed-forward network. Similarly, in the case of spatial attention of CBAM, an attention score is calculated using a 1D-CNN on the compressed result using average pooling and max pooling in the channel direction. In this study, the attention layer similar to the described CBAM model was applied to timeseries data. As an output, a new highlighted important feature could be acquired.

The channel attention layer of the proposed model is the same as that in CBAM as shown in the upper part of Figure 3. The input of the attention layer is compressed values of input data applying average pooling and max pooling to the direction of the time axis. Then, the attention map generation was achieved through the two types of inputs (average and max) passing through the same two 1D-CNN layers and the activation function, namely, rectified linear unit (RelU). The number of filters used in the first 1D-CNN of the channel attention layer was 1/16 of the number of input channels. Then, the number of filters used in the second 1D-CNN was recovered as the number of filters of input. This is the same as the channel attention layer described in the original CBAM, which increases the generalizing performance of the model, as explained in [22]. After the attention map generation, two attention maps are added element by element. Finally, through a sigmoid function that makes the values to be in the range of 0–1, the attention score is obtained. Then, the score is multiplied by input data for the channel axis. As a result, more important channels that better represent the data could be emphasized.

For attention at the spatial level, a self-attention layer is adopted. The original spatial attention layer presented in [22] is inappropriate for semantic segmentation in timeseries data because spatial information is greatly lost when average and max pooling are applied on the time axis. Therefore, the dot-product self-attention layer presented in [23], as described in the lower part of Figure 3, was used. There are three specific features—query, key, and value—that represent the input data differently. All features are generated through different 1D-CNN layers with the same input data; at this time, the same size as the number of frames of input data is maintained in the output. Then, the attention map is achieved by multiplying the query and key. The attention map of the original self-attention is a relation of each position. In timeseries, the relation of each time step is represented in an attention map. Then, the attention score is achieved using a sigmoid activation function similar to the channel attention layer. Finally, the output of the layer was derived by multiplying the attention score and the value representing the input data. This means that the features of each time step of the output are emphasized considering the entire data. This does not lose positional (time-domain) information and allows the model to be trained to recognize every frame without specifying the input data size. In other words, when calculating the features of a specific frame, the network can be trained to emphasize the features in positions important for classifying. Therefore, it is more suitable for performing semantic segmentation than the spatial attention layer in CBAM.

#### 2.1.3. Semantic Segmentation and Loss Function

The proposed model is designed by stacking three structures comprising a feature extraction layer and an attention layer. In addition, the output of each layer maintains the size of data for the time axis the same as the initial input data. Thus, if necessary, zero-padding is applied to the input data. As mentioned above, semantic segmentation in a 2D image means classifying every pixel in the image. In this study, to apply this approach to timeseries, the feature size of the final output of the model was matched with the size of the target classes to be predicted. For implementation, the output was fed into a fully connected layer with the same filter size as the target classes. Finally, the features corresponding to each frame pass through the softmax function to generate a probability distribution and are encoded with the value of the position with the maximum probability. To train the proposed DL model, the cross-entropy loss, a loss function mainly used in classification problems, is applied in every frame. The losses generated in each frame are summed up as the final loss value of training the proposed model, as described in Formula (1). In the actual training phase, the number of filters used in every block and layer was 64.
(1)$$Total\;Cross\;Entropy\;Loss=-\overset{T}{\underset{j=1}{\sum }}\overset{C}{\underset{i=1}{\sum }}y_{j}^{i}\log\left(z_{j}^{i}\right),$$
where T denotes the length of the time axis of input data, C denotes the number of classes, $y_{j}^{i}$ denotes the true label of data at time j, and $z_{j}^{i}$ denotes the probability from the softmax function of the recognition model for class i at time j.

### 2.2. Dataset Construction

There are various human activity datasets comprising acceleration data, such as WISDM, UCI HAR, and MHEALTH [24], but most of them focus on the change in the human state, not on the transition activity. The human state is changed by transition activity and can mainly refer to human postures. For example, after a transition activity of sitting, the human state becomes seated. However, to precisely recognize the target activity, it is important to recognize the transition activity in which the target state is transformed. In addition, if the transition activity can be recognized with high accuracy, the human state can be predicted easily. Nevertheless, most public datasets are labeled the same for the transition activity and subsequent target state. In other words, there is no distinguished label between the state and transition activity. Therefore, a new dataset comprising only types of transition activity was constructed in this study.

As previously mentioned, there are various issues in public datasets for implementing and evaluating semantic segmentation for human activity data. Therefore, a new dataset was constructed using watch type IMU. The target activities comprise get-up, laying, stand-up, picking, sitting, and walking. Further, the background class means no movement is included. The target classes comprise behaviors that can occur in human daily life, which are commonly included in many public datasets. Data comprising two activities are included in the dataset. The two behavioral data types include all combinations that humans could perform in the target classes. The sensors used to construct the dataset consisted of a UWB sensor and an IMU consisting of a 3-axis (x, y, z-axis) accelerometer (LIS2DS12TR, STMicroelectronics), as shown in Figure 4. The UWB signal, which provides the location information of the indoor sensor, was not used in this study. However, it will be used for future studies that use context information to improve recognition performance. The acceleration data capturing speed was set to 15 fps. Therefore, the movement of the subject was captured with the sampling duration of 66.6 ms. All subjects wore the provided sensors on their right wrists and performed the motion for 250 frames. This means the size of a single sample (the input data) was 250 frames. Therefore, every target activity data were collected by all subjects, not only the single action, but also the two behavioral data (the combination of two action), has a size of 250 frames. Subjects were 8 males between the ages of 20 and 40 years. In addition, all subjects performed 6 single actions and 12 multi-actions, 10 times each, for a total of 180 times. All data were labeled with the corresponding activity by pinpointing the starting and ending points. The labeling procedure was performed manually by one person who watched all movements of all subjects. In addition, activities were performed at various time points in the single data. The state and target activities of the dataset are described in Table 2.

## 3. Results

Two experiments were performed to evaluate the proposed model on the new dataset. First, we evaluated the performance improvement compared with the basic DL modules and models proposed in the ECG segmentation studies. Second, we experimented to evaluate how the method of CEF proposed in this study is more accurate than SW. The evaluation metrics for all verification were the F1-score, precision, and recall:(2)$$Precision=\frac{TP}{TP+FP}$$
(3)$$Recall=\frac{TP}{TP+FN}\;\;$$
(4)$$F1=\frac{2\times Precision\times Recall}{Precision+Recall},$$
where TP (True Positive) means a result of predicting a class in which the predicted value for the data at a specific point in time is actually correct, FP (false positive) means that the predicted value of data that is not a specific class is recognized as that specific class, and FN (False Negative) means a result recognizing that data that are not actually of the corresponding class are of the corresponding class.

In addition, experiments were performed on an Intel i9-11900F octa-core microprocessor clocked at 2.50 GHz with 32 GB RAM. For operating the proposed DL model and all comparison models, the RTX 3070 GPU was used. Model development and implementation were performed using Pytorch version 1.10.2 and Python version 3.7, respectively. The size designed model was 13.063 MB with 3,416,583 parameters. All models were trained using a leave-one-subject-out cross-validation, with the number of epochs fixed at 200 during each training. The learning rate and batch size for learning were set to 0.001 and 100 in all experiments, and the Adam method was employed for optimization. The experimental results are described below, and the analysis is performed in the next section.

In the first experiment, the basic DL modules used for comparison included the gated recurrent unit (GRU), LSTM, and CNN modules with different kernel sizes. For RNNs, bidirectional modules were also employed for comparison, and kernel sizes of 5, 10, 20, and 40 steps were used for CNN. All basic DL modules were stacked three times, including batch normalization and ReLu activation function; finally, an output having the same feature size as the input data was obtained through a fully connected layer. In addition, several models presented in [25,26,27,28,29] that reported high accuracy by applying the CEF method to ECG data were adopted for comparison. As mentioned in Section 2, the loss is calculated by comparing the output of all models with the same size as the input and the label on the time axis. Details of the results for each activity are presented in Table 3. Additionally, the computational cost of all models (the number of parameters, and the size of the model) are described in Appendix A.

The proposed model reported the highest performance in all classes, with an F1-score of 0.9 or higher. In detail, for the background, the proposed model reported the best F1-score and precision values of 0.979 and 0.978, respectively. However, for the recall value, [29] was the best, with 0.990. Considering laying, picking, get-up, stand-up, and sitting, for precision, the results of [29] were the best, with 0.945, 0.936, 0.948, 0.91, and 0.94; meanwhile, for recall, the results of the proposed model were the best, with 0.927, 0.930, 0.930, 0.896, and 0.905, respectively. For walking, the CNNs with kernel sizes of 20 and 40 steps had the highest precision, 0.950, but the recall of the proposed model was the highest (0.953). Overall, the F1-score of the proposed model was the best, with an average of 0.929, and it was 0.019 higher than that of the CNN with a 40 step kernel size, which is the highest F1-score among the comparison models. The confusion matrix of experiments is provided in Appendix A.

In the second experiment, the comparison models were the same as in the first experiment. However, to reproduce the SW method, the dataset comprising 250 frames for single data was divided into several segments according to the window size and the size of the overlapping area. Then, all recognition models classified the segment as a specific behavior. As a result, the output data differ from the input data and refer to the class of corresponding data that the recognition model predicted. For predicting overlapping areas, the class is determined by comparing the confidence between surrounding predictions. In other words, the prediction which has a higher confidence value from the same recognition model in different positions is selected as the final decision. For an accurate comparison, the predicted class is expanded by its original size, and the loss is calculated through comparison with the label. For training models, the aforementioned cross-entropy was adopted as the loss function. The various sizes of the fixed window were set to 10, 20, and 40 time steps, and each overlapping area had a size of 5, 10, and 20 steps. In addition, the evaluation criterion is the averaged F1-score of all activities. Details of the results are described in Table 4.

The proposed CEF method reported the best performance compared with three types of SW. The F1-score of SW with a window size of 10 and an overlapping area of 5 was 0.732. For SW with a window size of 20 and an overlapping area of 10, the F1-score was 0.701. In addition, the F1-score of SW with a window size of 40 and an overlapping area of 20 was 0.742. Finally, the proposed CEF method showed the best performance with a 0.88 F1-score, an improvement of 0.154 on average. In detail, the results of CEF showed 0.198, 0.17, and 0.147 improvements compared with GRU, LSTM, and RNN, respectively. Meanwhile, the results of bidirectional RNN showed improvements of 0.171, 0.15, and 0.154 compared with GRU, LSTM, and RNN, respectively. Compared with CNN, the proposed CEF method showed improvements of 0.101, 0.14, 0.121, and 0.165 for kernel sizes of 5, 10, 20, and 40, respectively. Finally, the F1-score of CEF was higher than that of SW by as much as 0.184.

## 4. Discussion

In this study, the performance of a model designed by stacking various basic DL modules was evaluated through the first experiment mentioned in Section 3. For RNN modules, we confirmed that the average F1-score was 0.083 lower than that of the bidirectional RNN using not only the previous but also the next information. RNN modules may be unsuitable for predicting behaviors with long execution times due to problems such as gradient vanishing. Further, because it is difficult to apply bidirectional RNN in real time, we judged that the CNN module is more suitable for implementing CEF. In addition, among the CNN modules, because the one with the largest kernel size has a wide receptive field, the performance is the best among basic DL models. This means that using more than one piece of surrounding information to classify a particular frame has an advantage over RNNs. We also confirmed that the size of the receptive field used when classifying a specific frame greatly affects the performance. Moreover, depending on the execution time of the target action and the amplitude of the signal, features extracted from receptive fields of different sizes can have a positive effect on the prediction performance because the proposed model using MCNN had the best F1-score.

The model presented in [29], which reported the highest precision value in transition activities of laying, get-up, stand-up, sitting, and picking that have short execution times, used both 1D-CNN and dilated 1D-CNN for feature extraction. This means that the features from multiple receptive fields of various sizes had a positive effect on recognition performance. Therefore, when classifying data at a specific location on the time axis, it is essential to fuse meaningful features using surrounding data of various sizes. In the results of the CNN module with a single kernel size, the F1-score of transition activities with short execution time was 0.094 lower than the activities that include repetitive patterns, such as background and walking. This is because the information that interferes with predicting the class of a specific frame is included in the process of passing the features extracted from the previous layer to the next layer, and it can be improved by selectively filtering the necessary features. The model based on an autoencoder presented in [25,27] can compress and remove relatively insignificant features, but it has a loss of positional information, resulting in a low F1-score of 0.7. In addition, Refs. [28,29], which adopted a U-net architecture with a skip connection to preserve positional information, reported a relatively high F1-score of 0.757, but its performance was still low. Moreover, the model of [25], which emphasizes meaningful features by applying an attention layer that can give low weight to insignificant features, reported higher performance than the aforementioned two methods (0.874).

Consequently, the proposed model was designed by stacking MCNN that can reflect receptive fields of various sizes and an attention layer that emphasizes meaningful features. The attention scores of the channel level were derived differently for each activity (Figure 5). In other words, the features obtained from different sizes of the receptive field were emphasized selectively according to the properties of the target activities. In addition, the channel attention layer was applied differently according to the execution time of the target activity. When the execution time of an action was long, features extracted by kernels of all sizes were evenly emphasized; meanwhile, when the execution time was short, features extracted from a receptive field of a short size were emphasized. For spatial attention, the area of the same behavior as the data of a specific time step to be recognized was emphasized (Figure 6). This improved the classification performance of data at a specific location by filtering the data that are not related to the target and helped demarcate the boundary between the background and activity or distinguish between different adjacent activities. In summary, the proposed model was designed as a stacked structure by the fusion of the two methods, and it reported the highest performance.

In the second experiment, the proposed CEF was evaluated by comparison with the SW method. As a result, the performance of SW was lower than that of CEF because of several reasons. First, if the window size is too small or too large, a recognition error occurs. When a small amount of data is included, it may be insufficient to classify the data in the window. However, if a large amount of data is included in a single window, more than one activity may be involved, increasing the error. In other words, if a window contains more than one action, a misrecognition occurs, and it is impossible to specify the dividing point between the different actions. These problems can occur randomly according to the start and end points of the target behavior and recognition. From Figure 7, as a result of the SW method implemented in this study, misrecognition frequently occurred in the area corresponding to the division points, such as the start, end, and transition of the activities. Notably, the existing research treats the segment label as one class rather than a frame unit. This can have a positive effect on the recognition model’s training, but cannot perform quantitative evaluation precisely. Through additional tuning work, the performance of SW can be improved by obtaining the optimal window size and size of the overlapping area. However, performance improvements are not guaranteed for data with different behaviors or other datasets, because SW fundamentally depends only on the data contained in the window. Therefore, CEF, which is not limited by changes in window size, could perform activity recognition more precisely. In addition, by classifying each frame rather than the window unit, the distinction between various activities could be recognized more elaborately.

In this study, the CEF method was proposed to overcome the limitations of SW. However, several limitations still remain. First, as mentioned above, a CNN module with a different kernel size is required depending on the properties of the target activity. Therefore, the proposed model used features extracted from various receptive fields. Nevertheless, if more complex behaviors that are difficult to distinguish from other activities need to be recognized, a different kernel size may be more suitable. Thus, the number of CNN modules and the kernel size of the currently designed model have to be set up experimentally. Second, for spatial attention, where the attention layer is applied to time-axis data, the attention map size for calculating the score increases as the input data size increases. This issue can cause limitations when applying the recognizing model to embedded systems. Third, in this study, a quantitative comparison with previous studies was not performed. As mentioned in Section 2, there is a limitation to using the public datasets used in the previous SW-based HAR research, and the issue is that the evaluation criterion of SW differs from that of CEF, but various quantitative comparisons with state-of-the-art studies are needed. Finally, if the wearing position of the sensor is changed, the recognition performance may decrease. Therefore, there is a need for a method that can respond to various structures of sensors for generalization.

## 5. Conclusions

In this study, a CEF method, rather than the conventional SW method, was proposed for HAR. For implementation, features extracted from various receptive fields were used and fused using MCNN. Moreover, we could selectively weight the extracted features by proposing a layer that applies the attention mechanism to each channel and spatial level similar to CBAM. The channel level has the same structure as that of CBAM. Meanwhile, for the spatial level, a dot-product self-attention layer that does not lose positional information was adopted. Further, the proposed recognition model was evaluated using a newly constructed dataset. As a result, the proposed recognition model reported a higher F1-score than the models using basic DL modules. In addition, the proposed model outperformed several models applying CEF to EGC data. An experiment was also performed to verify the superiority of the proposed method over the existing SW method. It was found that the CEF method can perform HAR more precisely than the SW method with three different window sizes and overlapping areas. In addition, we confirmed that the proposed model is suitable for implementing CEF for HAR.

The performance of the proposed CEF method and recognition model was verified through experiments. However, several issues remain to be resolved, necessitating further studies for improvement. First, the proposed model will be advanced using a DL method that is more suitable to timeseries data, such as the temporal convolutional network (TCN) structure presented in [30]. We expect that the performance will be increased because the TCN that had a great performance in timeseries data could better memorize long-term memory for the time axis. In addition, it is expected that memory usage, which increases according to the input data size in spatial attention, can be reduced. The second is the design of a canonical domain transformer layer to increase the generalization performance of the proposed model. It is possible to increase the generalization performance by simply increasing the diversity of the dataset using various augmentation methods, but it is not a fundamental solution and requires a lot of time. Therefore, a layer that transforms the input data or extracted feature into a domain advantageous for recognition is required. A method such as the canonical domain transformer model suggested in [31,32] for obtaining a transformation matrix based on input data will be adopted to improve the generalization performance. Finally, the positional and historical context information will be used to improve recognition performance. The positional context information can be extracted using the distance between the target position and surrounding objects, such as bed, chair, and desk. For instance, when the target is close to the bed, the static activities related to bed such as laying can be more natural than dynamic actions such as running. These rules correspond to the positional context information and will be used in the learning process of an adapted model. To facilitate the collection of positional context information, a UWB sensor that can be attached to various objects to provide the positional information of the sensor of the targets in real time will be used. Meanwhile, historical context information can be extracted from actions that a subject has performed before. For example, after the subject performs the laying action, the subject cannot walk without stand-up or get-up action. Such constraining will enable training a recognition model to reduce the weight of activities inappropriate for the situation. Through these additional studies, if the generalization performance of the proposed CEF method can be improved, it will contribute to the research field of healthcare technology that requires precise recognition, such as a human monitoring system or elderly home care. Further, the recognized human’s daily activity will be a significant controlling factor for the proactive service of robots.

## Figures and Tables

**Figure 1 sensors-23-02278-f001:**
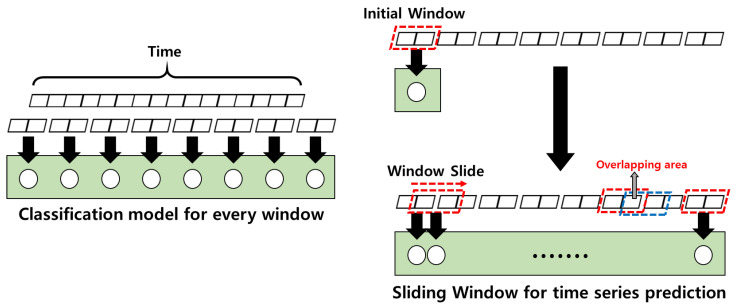
(**Left**) fixed-size window and (**Right**) sliding window for activity recognition.

**Figure 2 sensors-23-02278-f002:**
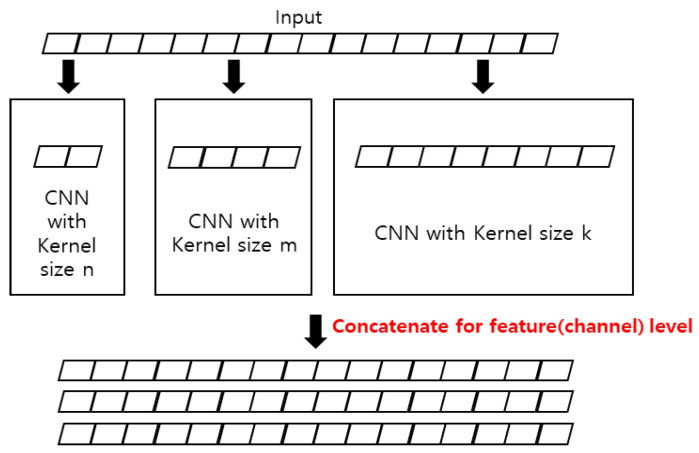
Feature extraction block using MCNN.

**Figure 3 sensors-23-02278-f003:**
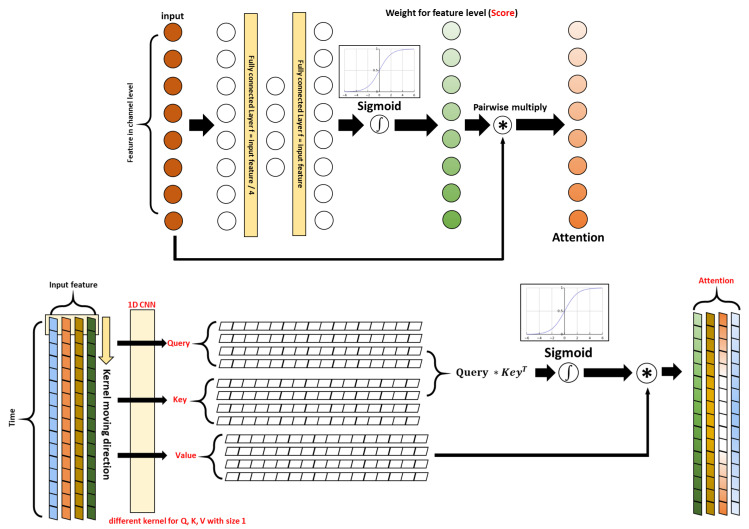
Designed CBAM (**up**) channel level and (**down**) spatial level.

**Figure 4 sensors-23-02278-f004:**
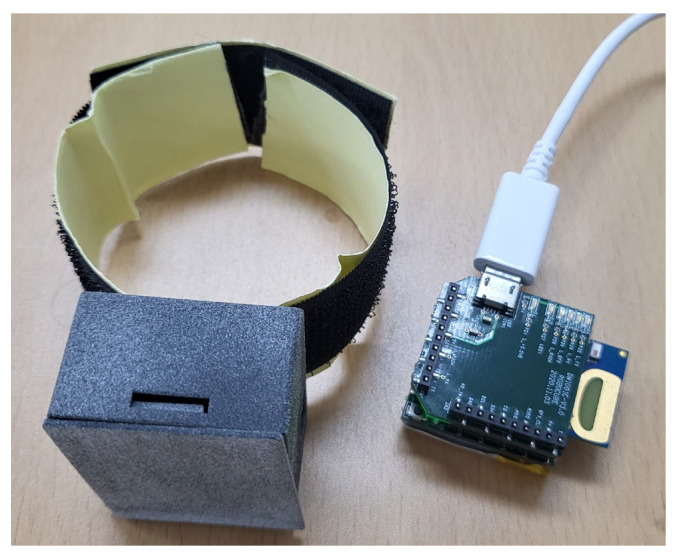
IMU sensor.

**Figure 5 sensors-23-02278-f005:**
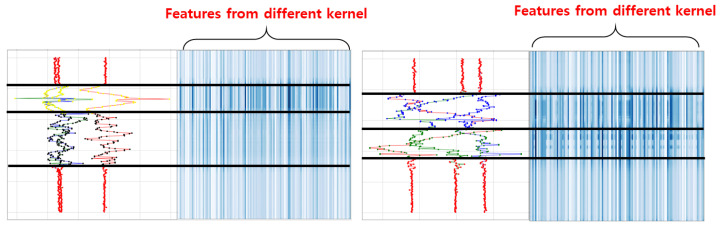
Examples of attention score in channel level.

**Figure 6 sensors-23-02278-f006:**
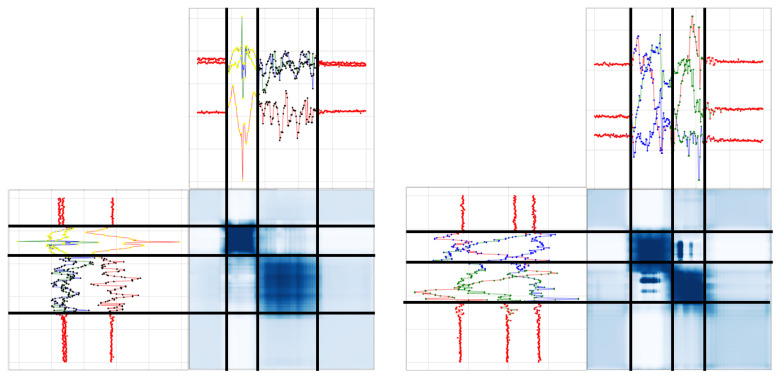
Examples of attention score in spatial level.

**Figure 7 sensors-23-02278-f007:**
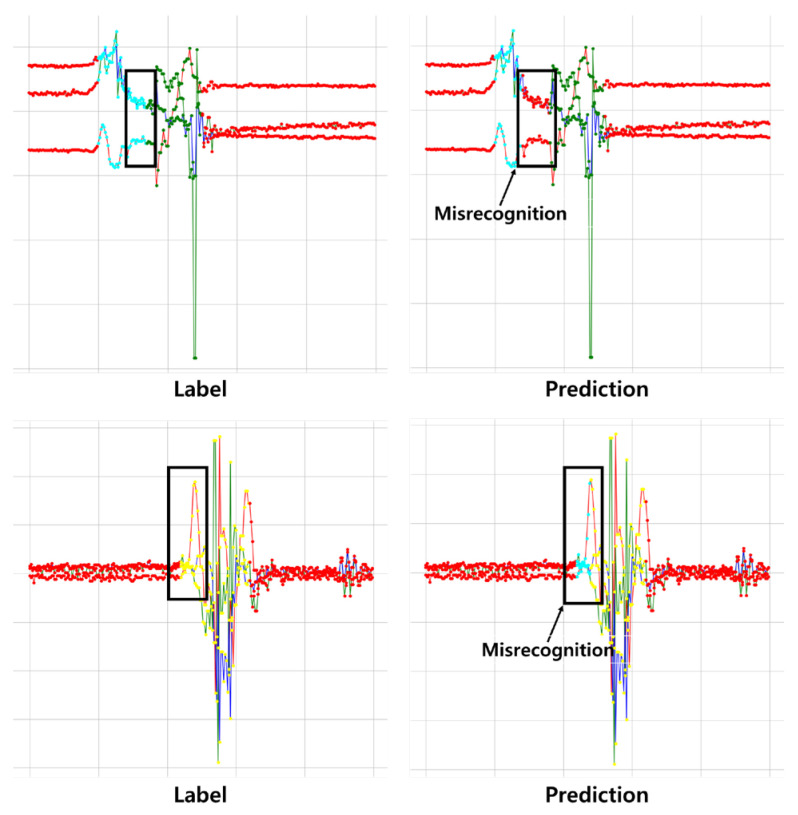
Example of misrecognition of SW method.

**Table 1 sensors-23-02278-t001:** Key characteristics of previous works.

Author	Dataset	Window Size (s)	Sliding Window (%)	Accuracy (%)	Model
[7]	[8]	0.7	50	92.22	1D-CNN
	[9]	3	78	93.68	1D-CNN
	[10]	1	99	70.80	1D-CNN
[11]	[8]	64	50	76.83	1D-CNN
	[12]	64	50	88.19	1D-CNN
	[13]	64	50	96.88	1D-CNN
[14]	[8]	500	50	94.2	1D-CNN
	[15]	500	50	97.62	1D-CNN
[16]	[8]	1	50	0.929 (F1-Score)	CNN + LSTM
	[9]	5.12	78	0.745 (F1-Score)	CNN + LSTM
	[17]	1	50	0.76 (F1-Score)	CNN + LSTM
[18]	[19]	10	90	94	LSTM
[20]	[8]	2.56	50	0.905 (F1-Score)	LSTM
	[15]	2.56	50	0.935 (F1-Score)	LSTM

**Table 2 sensors-23-02278-t002:** Status of constructed dataset.

ID	a	b	c	d	e	f	g	h
get-up	10	10	10	10	10	10	10	10
laying	10	10	10	10	10	10	10	10
stand-up	10	10	10	10	10	10	10	10
picking	10	10	10	10	10	10	10	10
sitting	10	10	10	10	10	10	10	10
walking	10	10	10	10	10	10	10	10
walking—picking	10	10	10	10	10	10	10	10
walking—sitting	10	10	10	10	10	10	10	10
stand-up—walking	10	10	10	10	10	10	10	10
sitting—laying	10	10	10	10	10	10	10	10
get-up—stand-up	10	10	10	10	10	10	10	10
picking—walking	10	10	10	10	10	10	10	10
get-up—laying	10	10	10	10	10	10	10	10
laying—get-up	10	10	10	10	10	10	10	10
stand-up—picking	10	10	10	10	10	10	10	10
stand-up—sitting	10	10	10	10	10	10	10	10
picking—sitting	10	10	10	10	10	10	10	10
sitting—stand-up	10	10	10	10	10	10	10	10

**Table 3 sensors-23-02278-t003:** Comparison with other DL models for evaluating performance improvement.

Background				Laying			Picking			Get-Up		
	Precision	Recall	F1-Score	Precision	Recall	F1-Score	Precision	Recall	F1-Score	Precision	Recall	F1-Score
GRU	0.960	0.964	0.962	0.815	0.864	0.839	0.808	0.838	0.822	0.880	0.811	0.844
LSTM	0.958	0.967	0.962	0.850	0.849	0.850	0.824	0.818	0.821	0.890	0.804	0.845
RNN	0.950	0.964	0.957	0.817	0.778	0.797	0.770	0.731	0.750	0.781	0.776	0.778
Bi RNN	0.966	0.974	0.970	0.878	0.885	0.881	0.886	0.875	0.881	0.884	0.832	0.857
Bi LSTM	0.973	0.971	0.972	0.874	0.904	0.889	0.882	0.860	0.871	0.870	0.872	0.871
Bi GRU	0.971	0.974	0.972	0.890	0.901	0.895	0.895	0.885	0.890	0.894	0.872	0.883
CNN 5	0.960	0.975	0.968	0.800	0.769	0.784	0.779	0.754	0.766	0.812	0.750	0.780
CNN 10	0.967	0.979	0.973	0.861	0.849	0.855	0.856	0.867	0.861	0.882	0.834	0.857
CNN 20	0.966	0.982	0.974	0.909	0.858	0.882	0.898	0.890	0.894	0.896	0.883	0.889
CNN 40	0.966	0.981	0.973	0.907	0.893	0.900	0.908	0.861	0.884	0.907	0.894	0.900
[25]	0.962	0.973	0.968	0.882	0.807	0.843	0.815	0.854	0.834	0.881	0.805	0.841
[26]	0.957	0.967	0.962	0.828	0.872	0.85	0.792	0.713	0.75	0.837	0.848	0.843
[27]	0.939	0.803	0.866	0.582	0.561	0.571	0.403	0.502	0.447	0.594	0.66	0.625
[28]	0.917	0.974	0.944	0.759	0.273	0.401	0.695	0.698	0.696	0.677	0.605	0.639
[29]	0.931	**0.99**	0.959	**0.945**	0.769	0.848	**0.936**	0.779	0.851	**0.948**	0.752	0.839
Proposed	**0.979**	0.977	**0.978**	0.918	**0.927**	**0.922**	0.913	**0.930**	**0.922**	0.909	**0.930**	**0.920**
**Stand-Up**					**Siting**				**Walking**			
	**Precision**		**Recall**	**F1-Score**	**Precision**	**Recall**	**F1-Score**		**Precision**	**Recall**		**F1-Score**
**GRU**	0.871		0.763	0.813	0.873	0.855	0.864		0.884	0.912		0.898
**LSTM**	0.843		0.782	0.811	0.841	0.836	0.838		0.875	0.904		0.889
**RNN**	0.806		0.690	0.744	0.816	0.767	0.791		0.835	0.885		0.860
**Bi RNN**	0.891		0.845	0.867	0.876	0.866	0.871		0.928	0.935		0.931
**Bi LSTM**	0.860		0.871	0.866	0.900	0.892	0.896		0.926	0.934		0.930
**Bi GRU**	0.892		0.871	0.882	0.901	0.885	0.893		0.936	0.951		0.943
**CNN 5**	0.804		0.739	0.770	0.812	0.765	0.788		0.880	0.913		0.896
**CNN 10**	0.876		0.847	0.861	0.885	0.851	0.868		0.943	0.927		0.935
**CNN 20**	0.905		0.867	0.886	0.914	0.872	0.893		**0.950**	0.929		0.939
**CNN 40**	0.906		0.863	0.884	0.912	0.879	0.895		**0.950**	0.928		0.939
**[25]**	0.858		0.839	0.848	0.877	0.853	0.865		0.914	0.925		0.92
**[26]**	0.786		0.771	0.779	0.759	0.795	0.776		0.892	0.832		0.861
**[27]**	0.432		0.368	0.397	0.465	0.356	0.403		0.276	0.559		0.369
**[28]**	0.715		0.416	0.526	0.693	0.444	0.541		0.652	0.886		0.752
**[29]**	**0.91**		0.801	0.852	**0.94**	0.743	0.83		0.902	0.936		0.918
**Proposed**	0.906		**0.896**	**0.901**	0.915	**0.905**	**0.910**		**0.950**	**0.953**		**0.952**

**Table 4 sensors-23-02278-t004:** Results of comparison with the SW method.

Model	GRU	LSTM	RNN	Bi RNN	Bi LSTM	Bi GRU	CNN 5	CNN 10	CNN 20	CNN 40	Proposed
SW 10-5	0.725	0.734	0.685	0.734	0.74	0.747	0.737	0.741	0.766	0.723	0.727
SW 20-10	0.631	0.657	0.65	0.753	0.725	0.717	0.672	0.701	0.781	0.718	0.71
SW 40-20	0.639	0.671	0.65	0.678	0.779	0.795	0.746	0.799	0.817	0.797	0.799
CEF	0.863	0.858	0.809	0.893	0.898	0.907	0.82	0.887	0.909	0.911	0.93

## Data Availability

Not applicable.

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
