# Peer review of "A Deep Learning-Based Semantic Segmentation Model Using MCNN and Attention Layer for Human Activity Recognition"

_sensors, 2023, doi:10.3390/s23042278_

Round 1

Reviewer 1 Report

This article provides a well documented and throughly valid scientific rationale, method, results, and review of actions needed to advance the use of digital measurements and human behavior recognition.

Development of methods for classification of movement behaviors using various methods of frame analyses and assessments.

The need to better segment and classify signals derived from human activity is a crucial area of research and this article provides useful insights and proposed methods.

This article evaluates methods of varying the frame, interval window, and characteristics of HAR and provides highly useful depiction of results from these activities.

This article is well balanced and appropriately detailed and does not require changes.

The conclusions and outcomes presented are detailed, thorough, related to the main question, and highly useful.

The references cited are appropriate and sufficient.

Reviewer 2 Report

The authors proposed a frame-based classification method using features extracted from multiple convolutional neural networks with varying kernel sizes, fused with an attention layer for improved recognition performance. Experiments reported good F1-score for all target activities on recognition of electrocardiography data. The method also has higher precision than existing sliding window methods in recognizing human activity.

The data collection process is unclear and requires further details, such as the sample size, participant count, sampling duration, and demographic information such as age and gender distribution. It is also unclear whether the actions are simulated in a laboratory or performed in real-life scenarios. The methods and comparative tests appear to be adequate, but without further information on the data collection, a complete analysis cannot be conducted. A section must be added to detail the data collection.

Figure 4 lacks annotations on the sensor and becomes ineffective. The sensor information missing from line 312 to 317, such as the model number, manufacturer, and a reference link to the datasheet, makes it difficult for readers to verify specifications and accuracy data of the sensor used. These details must be added. The authors should annotate Figure 4 to make it more useful to the reader.

Reviewer 3 Report

This paper propose a method for classifying all frames rather than a window-based recognition method for human activity recognition. The paper is written well and in organised way.

The following are the observation and required minor changes in this manuscript:

1. Abstact must contain the important finding of the manuscript in terms of results.

2.  The paper is written well. But some more performance matrics can be included for verifying the results, like Confusion matrix.

3. Overall good paper and is of public interest.

Reviewer 4 Report

The proposed method does improve overall performance. However, if possible, please provide comparison on computational cost theoretically or experimentally.

Reviewer 5 Report

Strengths:

 (+) The literature review is good.

 (+) The references are appropriate.

 (+) The experiments can be replicated. 

 (+)

 (+)

Weaknesses:

 (-) There are English issues.

 (-) The related work section must be enhanced.

 (-) Experimental evaluation must be improved.

 (-) Some improvements are needed in the description of the method.

 (-)

 (-)

==== ENGLISH ==== 

The writing style of this paper is not good enough. Authors should spend some time improving it so that the paper can be read more smoothly.

==== RELATED WORK ==== 

The authors should add a table that compares the key characteristics of prior work to highlight their differences and limitations. The authors may also consider adding a line in the table to describe the proposed solution.

==== METHOD ==== 

The authors should first give an overview of their solution before explaining the details. 

A novel solution is presented but it is important to better explain the design decisions (e.g. why the solution is designed like that)

The solution is described but there should be more examples.

==== EXPERIMENTS ==== 

The experiments should be updated to include some comparison with newer studies. 

A statistical analysis should be carried out to demonstrate that the experimental results are significant. 

Some additional experiments are required:

 - Scalability

 - Memory

==== CONCLUSION ==== 

Some text must be added to discuss the future work or research opportunities.

Round 2

Reviewer 2 Report

Thank you for accepting the feedback and revising the manuscript accordingly. 

Reviewer 5 Report

All the earlier concerns have been properly addressed.